# Calcium-Activated Big-Conductance (BK) Potassium Channels Traffic through Nuclear Envelopes into Kinocilia in Ray Electrosensory Cells

**DOI:** 10.3390/cells12172125

**Published:** 2023-08-22

**Authors:** Abby L. Chen, Ting-Hsuan Wu, Lingfang Shi, William T. Clusin, Peter N. Kao

**Affiliations:** 1Department of Medicine, Division of Pulmonary, Allergy and Critical Care Medicine, Stanford University School of Medicine, Stanford, CA 94305, USA; achen95@stanford.edu (A.L.C.); thsuanwu@stanford.edu (T.-H.W.); lfshi@stanford.edu (L.S.); 2Department of Biochemistry, Stanford University School of Medicine, Stanford, CA 94305, USA; 3Department of Medicine, Division of Cardiovascular Medicine, Stanford University School of Medicine, Stanford, CA 94305, USA; wclusin@stanford.edu

**Keywords:** *Leucoraja erinacea*, ampullae of Lorenzini, electroreception, nuclear localization, alternative splicing, STREX exon, mutagenesis

## Abstract

Electroreception through ampullae of Lorenzini in the little skate, *Leucoraja erinacea*, involves functional coupling between voltage-activated calcium channels (CaV1.3, *cacna1d*) and calcium-activated big-conductance potassium (BK) channels (BK, *kcnma1*). Whole-mount confocal microscopy was used to characterize the pleiotropic expression of BK and CaV1.3 in intact ampullae. BK and CaV1.3 are co-expressed in electrosensory cell plasma membranes, nuclear envelopes and kinocilia. Nuclear localization sequences (NLS) were predicted in BK and CaV1.3 by bioinformatic sequence analyses. The BK NLS is bipartite, occurs at an alternative splice site for the mammalian STREX exon and contains sequence targets for post-translational phosphorylation. Nuclear localization of skate BK channels was characterized in heterologously transfected HEK293 cells. Double-point mutations in the bipartite NLS (KR to AA or SVLS to AVLA) independently attenuated BK channel nuclear localization. These findings support the concept that BK partitioning between the electrosensory cell plasma membrane, nucleus and kinocilium may be regulated through a newly identified bipartite NLS.

## 1. Introduction

Elasmobranchii are a subclass of cartilaginous fish capable of detecting nanovolt gradients in seawater through electrosensory ampullae of Lorenzini [1,2]. Epidermal pores clustered around the mouth and gills open into canals filled with electroconductive jelly that terminate in the lumens of ampullary sensory epithelia [3,4,5]. The single-cell-thick epithelium comprises electrosensory cells with kinocilia facing the lumen at the apical surface and a ribbon synapse that connects to the sensory nerve at the basal surface.

Electroreception involves the interplay between a depolarizing inward calcium current and a hyperpolarizing calcium-activated outward potassium current [6,7,8]. Large-conductance calcium- and voltage-activated potassium channels (BK) are ubiquitously expressed (reviewed in [9,10]). We were the first to clone the cDNA encoding the skate BK channel based on its homology to the mouse Slo1 gene [11]. Bellono et al. performed a transcriptomic analysis of skate ampullae and assembled cDNA sequences for highly expressed CaV 1.3 (*cacna1d*) voltage-gated calcium channels and BK (*kcnma1*) calcium-activated potassium channels [12]. They demonstrated by in situ hybridization the colocalization of CaV1.3 and BK RNAs within skate ampullary electrosensory cells.

Alternatively spliced isoforms of BK channels are trafficked to plasma membranes, nuclear membranes or mitochondria [13,14,15,16,17,18,19,20,21,22]. BK channels assemble into macromolecular complexes that frequently include voltage-activated calcium channels [9,23]. Functional interactions and proximal clustering between BK and CaV1.3 channel proteins in plasma membranes of rat hippocampal neurons were characterized by electrophysiology and super-resolution microscopy [24].

Sensory cilia serve as antennae to increase sensitivity for environmental signals (reviewed in [25,26]). Because the volume within a cilium is several-thousand-fold smaller than the cytoplasm of the cell body, voltage gradients will trigger ion fluxes with a higher likelihood of crossing activation thresholds in cilia than in cell bodies.

Here, we perform whole-mount confocal microscopy on intact skate ampullae and describe the multifarious localizations of BK- and CaV1.3-channel proteins involved in electroreception. BK and CaV1.3 channels are expressed in electrosensory cell plasma membranes and prolifically in nuclear envelopes and kinocilia. Kinocilia appear to emerge from nuclear envelopes. We use bioinformatic sequence analysis to identify a new bipartite nuclear localization sequence and validate that it regulates trafficking of BK channels into nuclei in heterologous cells. To our knowledge, this is the first study to describe co-expression of BK, CaV1.3 and lamin B1 in nuclear envelopes and kinocilia of electrosensory cells. Ampullae of the little skate offer a tractable system to explore how the regulated partitioning of BK channels between plasma membranes, nuclear envelopes and kinocilia may modulate electrosensory cell excitability and synaptic and nuclear signaling.

## 2. Materials and Methods

### 2.1. Ampullae Dissection and Whole-Mount Immunostaining

This study was carried out in strict accordance with the recommendations in the Guide for the Care and Use of Laboratory Animals of the National Institutes of Health. The protocol was approved by the Administrative Panel on Laboratory Animal Care of Stanford University (APLAC-26931). Adult skates (*Leucoraja erinacea*, Mitchill, 1825) were purchased from Marine Biological Labs in Woods Hole, MA, and shipped alive in chilled seawater. Animals were killed instantly by pithing of the brain and spinal cord. The ampullae were dissected with scissors from behind the gill slits. Approximately 200 ampullae with a short section of ampullary canal and afferent nerve were dissected in PBS under a stereomicroscope and cleaned of connective tissue. Ampulla were fixed in 1% paraformaldehyde in PBS for 1 h, then rinsed two times in PBS and stored for up to 1 week.

Whole-mount staining was performed in 0.5 mL microfuge tubes in 100 microliters of solution, then blocked with 10% FBS and 1% Triton X-100 detergent. Staining was performed with mouse and rabbit primary antibodies at final concentrations of 4 to 20 µg/mL in 0.1% FBS and 0.1% TX-100 for 48 h, followed by 3 serial washes with PBS-0.1% TX-100. Secondary antibodies were applied at 1:100 dilution: anti-mouse IgG(H+L) or IgG2α Alexa 488 or anti-rabbit Alexa 594. Nuclei were counterstained with DAPI. 

Primary antibodies used were (1) anti-BKα KCNMA1 PA1-923 (Invitrogen/ThermoFisher, Carlsbad, CA, USA), a rabbit polyclonal IgG, raised against synthetic peptide of human KCNMA1: T_945_ ELVNDTNVQFLDQDDD_961_, skate epitope is identical at 17/18 positions, underlined: TELVNDSNVQFLDQDDD, (2) MaxiKα (B-1) sc-374142 (Santa Cruz Biotechnology, Santa Cruz, CA, USA), a mouse monoclonal IgG2b raised against immunogenic amino acids 937–1236 at C-terminus of human MaxiKα, (3) Anti-Slo1, clone L6/60 MABN70 (Millipore, Burlington, MA, USA), a mouse monoclonal IgG2α, raised against a recombinant immunogenic protein corresponding to segment S9–S10 of mouse Slo1 (NP_034740), (4) CaV1.3 cacna1d: LS-B4915 (LifeSpan Biosciences, Seattle, WA, USA), a mouse monoclonal IgG2α, raised against immunogenic fusion protein encoding amino acids 859–875 of rat CaV1.3 (Accession P27732): DNK------VTIDDYQEEAEDKD, skate epitope is conserved at 11 of 16 positions, underlined: DRKILTGTQVSIDD-Q—DEDKD, (5) Lamin B1 Ab16048 (AbCam, Waltham, MA, USA), raised against immunogenic peptide corresponding to mouse Lamin B1 amino acids 400–500 conjugated to keyhole limpet hemocyanin, and (6) Calnexin HPA009433 (Sigma, St. Louis, MO, USA) Atlas Prestige Antibodies, a rabbit polyclonal IgG, https://www.proteinatlas.org/ENSG00000127022-CANX/antibody, accessed on 18 August 2023.

The specificity of anti-BKα KCNMA1 PA1-923 antibody was established by using blocking peptide ELVNDTNVQFLDQDDD (Genscript custom synthesis), 3 µg applied 2 h prior to primary antibody staining at 1 µg/100 µL (25-fold molar excess of blocking peptide). Secondary detection was performed at 1:200 dilution using AlexaFluor568 goat anti-rabbit IgG(H+L).

Confocal imaging was performed using Zeiss LSM 880 and Leica Stellaris 8 microscopes. Objective lenses used were apochromatic 10×/NA 0.45 and 40×/NA 1.2 glycerol immersion infinity/0.15–0.19 and 63×/NA 1.3 glycerol. Z-stacks of up to 100 microns were collected, using separate tracks to minimize interference between fluorophores. Image files were processed using Fiji (Image J2, Version 2.9.0/1.53t, NIH). Leica Stellaris z-stacks were acquired at the Nyquist limit and optically deconvolved using LIGHTNING software, 30 August 2020.

### 2.2. Site-Directed Mutagenesis

The Skate BK channel in mammalian expression vector pcDNA 3.1–kcnma1 was a generous gift from Bellono and Julius [12] (accession KY355737.1). Site-directed mutations were generated using an InFusion cloning system (ThermoFisher, Waltham, MA, USA) with designated primers (nucleotide changes from WT in capital letters): KCNMA1_NLSmuKR655AA_1954plus 5′-tgtgggtgtGCAGCAccaagatatggctataatggatatctcagcac, and KCNMA1_NLSmuKR655AA_1962minus 5′-GGTGCTGCacacccacatttcttaattctctttgtatctgtg; KCNMA1_NLS_SVLS674AVLA_2017plus 5′-ccaGcagtgctgGctcccaaaaaaaagcaacggaacgggggc, and KCNMA1_NLS_SVLS674AVLA_2033minus 5′-ggagCcagcactgCtgggttttcatcttgaattgtgctgag. Mutagenesis was confirmed by DNA sequencing. WT; KR → AA mutation and SVLS → AVLA mutations.

### 2.3. HEK293 Transfections and Immunofluorescence Staining

HEK293 cells were transfected on multiwell imaging slides. Between 300–700 ng of DNA plasmid was mixed with JetPrime lipid transfection agent and applied to cells at 50% confluency for ~40 h. Cells were fixed with 1% PFA and immunostained at 10 µg/mL using anti-BK mouse monoclonal anti-Slo1 (clone L6/60 MABN70) and rabbit polyclonal anti-lamin B1 (nuclear envelope marker) or anti-calnexin (ER marker [27]) antibodies. Nontransfected cells did not demonstrate prominent Slo1 staining. Transfected cells with prominent (intracellular) Slo1 staining were counted, and the fraction of those cells that exhibited overlap with DAPI nuclear stain was counted manually. The effect of site-directed mutations on nuclear localization of skate BK channels was determined quantitatively. For HEK293 cells transfected with Wild Type BK, we examined 9 fields containing 537 cells; for KR → AA mutation, we examined 7 fields containing 547 cells; and for SVLS → AVLA mutation, we examined 5 fields containing 285 cells. The fraction of bright-green cells (5–9%) with BK overlapping the nucleus (marked by DAPI) was determined. Because these fractions were skewed around the means, we used the Kruskal–Wallis statistic based on ranks to determine significance. The differences between WT and AA and between WT and AVLA mutations were statistically significant with *p* < 0.05. Statistical analysis was performed using Microsoft Excel.

## 3. Results

### 3.1. Ampulla Expression of BK, CaV1.3 and Lamin B1

We microdissected single ampullae and performed whole-mount immunostaining followed by in situ confocal microscopy. Ampullae open into six or seven alveoli, which consist of single-cell-thick epithelial electrosensory cells with apical kinocilia directed into the lumen [3,4,5]. We established the specificity of anti-BK*α* KCNMA1 PA1-923 by the addition of blocking peptide corresponding to the immunizing epitope, which completely abolished the BK immunoreactivity in ampullae (Figure 1, panel D vs. A).

A confocal section across the wall of an ampullary alveolus is shown in Figure 2, co-immunostained to reveal the expression of CaV1.3 and BK channels (Figure 2; Appendix A). The inset shows a magnified view of the lateral wall that highlights the transmural character of single epithelial electrosensory cells. CaV1.3 is expressed in the apical, lateral and basal plasma membranes, with the most prominent expression seen at the bases (labeled in Figure 2A, inset; Appendix A). There is very faint granular staining of CaV1.3 over the nuclei closest to the cell bases. BK expression is more heterogeneous than CaV1.3 in electrosensory cells. BK is expressed in apical, lateral and basal plasma membranes, visible as thin straight lines (labeled in Figure 2B, inset; Appendix A). BK is expressed prominently in nuclei, exhibiting a coarse, clumped pattern near the base and a fine, granular pattern near the apex of electrosensory cells (Figure 2B, inset; Appendix A). The nuclear expression of BK is established by colocalization with DAPI (Figure 2C,D, inset). We interpret the rod-like protrusions into the lumen that co-express CaV1.3 and BK to represent apical kinocilia (labeled in Figure 2A,B,D). We observed prominent nuclear asymmetry in electrosensory cells, marked by coarse, clumped DAPI staining closer to the base and fine, granular DAPI staining closer to the apex (labeled in Figure 2C, Appendix A), Some nuclei looked like twisted dumbbells, with basal and apical lobes exhibiting asymmetric staining by DAPI, BK and CaV1.3 (Figure 2; Appendix A).

We next investigated the localization of nuclear BK channels with respect to the nuclear envelope, using lamin B1 (LamB1) expression to mark the inner nuclear envelope [20]. A “dome” view of an alveolus co-immunostained with anti-BK and anti-LamB1 and counterstained with DAPI is shown in Figure 3 and Appendix A. The anti-BK mouse monoclonal antibody (MaxiKα) demonstrated BK expression in perinuclear rings (Figure 3A,C,D), which closely correlated with nuclear envelope staining by anti-LamB1 (Figure 3B). The merged image is consistent with the co-expression of BK and lamin B1 antigens within 0.1 µm (Figure 3D, yellow). We note fine, granular stippling and co-expression of BK and lamin B1 within cell nuclei, which coalesces into perinuclear envelope labeling (Figure 3A,B,D top right). Invaginations of the nuclear envelope into the nucleus form the nucleoplasmic reticulum, which contributes to ion transport, signal transduction and regulation of gene expression (reviewed in [28]). BK, CaV1.3 and lamin B1 are co-expressed in nuclear envelopes that surround the apical poles of electrosensory nuclei (Figure 2 and Figure 3). The dense puncta that emerge from the apical nuclear envelopes and co-express BK, CaV1.3 and lamin B1 are likely to represent the bases of sensory kinocilia (Figure 2 and Figure 3A,B,D, labeled). We present movies of confocal z-stacks through the ampullae that better convey the colocalization of BK and lamin B1 or CaV in rod-like structures, which we interpret to represent kinocilia. This may be best appreciated in Appendix A (merge of BK and lamin B1, colocalization in yellow) at 8 seconds (near the bottom, two kinocilia) and 14 seconds (at the top left, six kinocilia), shown as yellow rods in transverse or cross-section, respectively.

We observed the strong expression and colocalization of CaV1.3 and BK channels in sensory kinocilia (Figure 4; Appendix A). Furthermore, the kinocilium emerges from the underlying nucleus. Colocalization of BK and CaV in yellow rods emerging from nuclei may be appreciated in Appendix A at 2 s (middle left) and 3 s (upper right).

We previously demonstrated expression of lamin B1 in nuclear envelopes and kinocilia (Figure 3B; Appendix A); we conclude that the ciliary membrane is contiguous with the inner nuclear membrane. The proximity between BK and CaV1.3 channels will enable voltage-gated calcium influxes to activate functionally coupled BK channels and modulate activation near the threshold, as described [24]. Thus, regulated trafficking of BK and CaV1.3 channels into sensory kinocilia that results in high-level co-expression will enhance the sensitivity of electroreception.

### 3.2. BK Nuclear Localization Sequence Characterization

We analyzed the skate BK channel primary amino acid sequences for potential nuclear localization signals using the program cNLS mapper [29]. The King (KJ756351) [11] and Bellono (KY355737) [12] sequences both contain a bipartite NLS, with different intervening linkers (Table 1). Note that Bellono has a 12-amino-acid insert (underlined) that is unique to their sequence. These closely related, alternatively spliced bipartite NLS sequences are present at alternative splice site 4, also known as c2 [16]. A 59-amino-acid STREX exon alternatively spliced into this location [30] would substantially increase separation between the two clusters of basic lysines (K), potentially attenuating functionality of the NLS. A profile of the complex phosphorylation status of the BK alpha subunit in rat brain showed that the PPTLSP sequence, or PSTLSP in humans (NP_001309759.1), is frequently phosphorylated [31]. When we analyzed skate CaV1.3 (KY355736) [12] using cNLS mapper, we identified a monopartite NLS at position 1583: YFRKFKRRKEQ.

We next used BlastP to search across species for proteins with sequence homology to the skate BK (*kcnma1*) bipartite NLS (Appendix A). We found that the bipartite NLS with variable intervening linker in *kcnma1* is highly conserved across diverse species. Vertebrates contain two clusters of basic amino acids, KRIKKCGCKR* upstream (* marks alternative splice 4, also known as c2) and KKKQRNG downstream. These sequences are absent in invertebrates, although Drosophilas exhibit some homologous basic amino acids in these positions [14].

We evaluated this bipartite NLS for its ability to confer nuclear localization to skate BK channels transiently expressed in HEK293 cells. Skate *kcnma1* cDNA cloned into pcDNA3.1 mammalian expression vector was a gift from Nicholas Bellono and David Julius [12]. Site-directed mutations were generated within the bipartite NLS. In the proximal portion of the bipartite NLS, two basic amino acids were mutated to alanine, KR to AA. Separately, two serine residues within the linker domain that are sites of phosphorylation were mutated to alanine, SVLS to AVLA. HEK293 cells were either nontransfected or transiently transfected with *kcnma1* expression vectors for Wild Type, KR to AA or SVLS to AVLA mutations. After 40 h, HEK293 cells were fixed with paraformaldehyde, permeabilized with TX-100 and co-immunostained for BK channel expression (Slo1 antibody) together with subcellular markers for nuclear envelope, lamin B1 or endoplasmic reticulum and calnexin and counterstained with DAPI.

Compared to control cells, HEK293 cells transfected with BK channels exhibited increased BK immunoreactivity in plasma membranes and intracellular compartments (Figure 5F vs. Figure 5A, Figure 6F vs. Figure 6A). Approximately 5–10% of all transfected cells showed high levels of intracellular BK expression. HEK293 cells expressing WT BK channels showed substantial colocalization with DAPI and lamin B1, consistent with the expression in nuclei and nuclear envelopes (Figure 5F–J). In contrast, HEK293 cells transfected with KR to AA or SVLS to AVLA mutations showed a high expression of BK channels adjacent to but not overlapping nuclei stained with DAPI and lamin B1 (Figure 5K–T). We note that accumulation of perinuclear-excluded BK channels caused indentations of the nuclei (arrowheads in Figure 5L,Q and Figure 6L,Q).

The high intracellular expression of BK channels adjacent to nuclei suggested to us that BK channels were accumulating in endoplasmic reticulum (ER). Calnexin is an integral membrane protein chaperone that localizes to and is used as a marker of the ER [27]. We co-immunostained transfected HEK293 cells with BK and calnexin antibodies (Figure 6). HEK293 cells expressing WT BK channels showed higher levels of colocalization with DAPI than with calnexin, consistent with nuclear translocation (Figure 6F–J). In contrast, HEK293 cells transfected with KR to AA or SVLS to AVLA mutations showed substantial colocalization with calnexin, and not with DAPI, consistent with retention of BK channels in the ER and exclusion from nuclei (Figure 6K–T). Again, we noted that the accumulation of nuclear-excluded BK channels caused indentations of the nuclei (arrowheads in Figure 6L,Q).

We counted HEK293 cells with high intracellular expression of BK channels and determined the subcellular localization pattern in relationship to WT, KR to AA or SVLS to AVLA mutations (Figure 7). WT BK channels translocate into the nucleus at 69% frequency, whereas mutations KR to AA or SVLS to AVLA significantly reduced nuclear translocation to ~15%. Compared to WT-transfected cells, SVLS to AVLA mutant transfected cells showed higher expression of intracellular BK channels (~10% vs. ~5%) and greater survival after transient transfections.

We determined that skate *kcnma1* contains a functional bipartite NLS and identified specific amino acid mutations that serve to attenuate nuclear translocation.

## 4. Discussion

We studied the expression and localization of ion channels involved in electroreception through ampullae of Lorenzini of the little skate. The single-cell-thick epithelium in the alveoli is amenable to in situ confocal microscopy, which enabled us to localize voltage-activated calcium channel CaV1.3 and calcium-activated BK channels in electrosensory cell plasma membranes, nuclear membranes and kinocilia. Our experiments were enabled by Bellono et al., who first described the expression and electrophysiological properties of little skate CaV1.3 (*cacna1d*) and BK (*kcnma1*) [12]. Electrical coupling between CaV1.3 and BK channels underlies the generation of membrane voltage oscillations in electrosensory cells that control the synaptic release of neurotransmitter [6].

Proximal clustering that promotes functional coupling between CaV1.3 and BK channels to support activation at low voltage was previously described [23,24]. Vivas et al. employed super-resolution microscopy to demonstrate proximal clustering of CaV1.3 and BK channels within 50 nm in rat hippocampal neurons and in transfected tsA201 (transformed HEK293) cells. Clustering formed functional multichannel complexes of CaV1.3 and BK channels at variable stoichiometry that enabled BK channels to open at relatively negative voltages [24]. The resolution of our confocal study colocalizes BK and CaV1.3 channels within 100 nm in ampullae of little skates.

Intracellular BK channel localization has been described in nuclei, endoplasmic reticulum and mitochondria [13,17,18,21]. Singh et al. identified a 50-amino-acid C-terminal splice insert as essential for targeting of BK channels to mitochondria [32]. Li et al. described BK expression in nuclear envelopes that colocalized with lamin B1 in rat hippocampal neurons [20]. Functional BK channels in the inner nuclear membrane modulate calcium entry into the nucleus and calcium-activated gene transcription. We observe BK expression in a coarse nucleoplasmic reticulum [28] at the basal poles of electrosensory cell nuclei that transitions to circumferential nuclear envelope BK expression at the apical poles of nuclei closest to the lumen.

The trafficking of skate BK channels into nuclei and nuclear envelopes is regulated by a bipartite NLS, modulated by serine phosphorylation at an SVLS motif [31]. The bipartite NLS occurs across splice site 4 [16], also identified as c2 [15]. Insertion at this site of alternative exons, including the STREX exon, increases the separation between the two clusters of basic amino acids and may disrupt its function as an NLS. The “SRKR” exon insertion at splice site 1, modulated by circadian rhythm, affects plasma membrane BK channel activation [33]. This insertion adds three basic amino acids 84 residues upstream of the mouse kcnma1 bipartite NLS and creates an additional NLS that might affect nuclear trafficking; however, internalization of BK channels from plasma membranes to intracellular organelles was not characterized.

Membrane proteins such as BK channels may be transported to the inner nuclear membrane through interactions of internal NLS with karophyrins/importins (reviewed in [34]). Internalization of BK channels from plasma membranes, modulated by alternative splicing and reversible phosphorylation of the NLS, would reduce plasma membrane BK current and increase cell excitability. Such reduction in plasma membrane BK current is associated with increased excitability in late-pregnant rat myocytes [35]. In mice, decreased plasma membrane potassium currents corresponded with increased electrical excitability of late-pregnant myometrium, and this was associated with internalization and perinuclear accumulation of BK channels [36].

Hutchinson–Gilford progeria syndrome (HGPS) manifests as premature aging and is caused by genetic mutations in lamin A that lead to the progressive breakdown of the nuclear envelope [37]. Zironi et al. demonstrated BK channel overexpression in plasma membranes by electrophysiology and immunofluorescence of dermal fibroblasts isolated from HGPS patients compared to young and elderly control donors [38]. These results are consistent with a model where functional BK channels that normally partition between plasma membranes and nuclear envelopes may exhibit inappropriately increased expression in plasma membranes when nuclear envelopes are genetically disrupted. Overexpression of BK channels in plasma membranes of HGPS patients might impair normal activation of pancreatic beta cells in response to glucose, conferring susceptibility to diabetes [39].

Kinocilia are specialized structures scaffolded by microtubules and enveloped by plasma membrane that contain extremely high densities of signaling receptors including ion channels [25,26]. The geometry of the cilium leads to a volume 5000-fold lower than the cytoplasm and enables signal amplification and quantal detection in sensory transduction as exemplified by rhodopsin expression in rod photoreceptors. Kinocilia are anchored to basal bodies derived from centrioles and assembled from nucleus-associated microtubule organizing centers, the centrosomes (reviewed in [40,41]). A basal body–nucleus connector is clearly seen in electron microscopy images of flagellar roots or rhizoplasts in the green alga, *Chlamydomonos reinhardtii* [42,43]. Here we demonstrate colocalization of CaV1.3 and BK channels together with lamin B1 in nuclear envelopes of electrosensory cells. Antenna-like kinocilia emerged from nuclear envelopes marked by high-density clustered expression of BK, CaV1.3 and lamin B1. Signals and mechanisms that control ciliary localization of BK and CaV1.3 channels remain to be elucidated. Deficiency in lamin A/C, either in LMNA-/- mice or in HGPS fibroblasts, was associated with disrupted integrity of the nuclear lamina and impaired ciliogenesis [44].

Skate ampullary electrosensory cells exhibit high intracellular expression of BK channels in nuclei. Our identification of a novel bipartite NLS modulated by alternative splicing or reversible phosphorylation suggests that BK channels can partition dynamically between plasma membranes, nuclear envelopes and kinocilia. BK channels in kinociliary and plasma membranes likely interact functionally with voltage-activated calcium channels to tune electroreception. BK channels in nuclear envelopes represent a reservoir for kinociliary and plasma membrane expression and may contribute to gene regulation [20].

## Figures and Tables

**Figure 1 cells-12-02125-f001:**
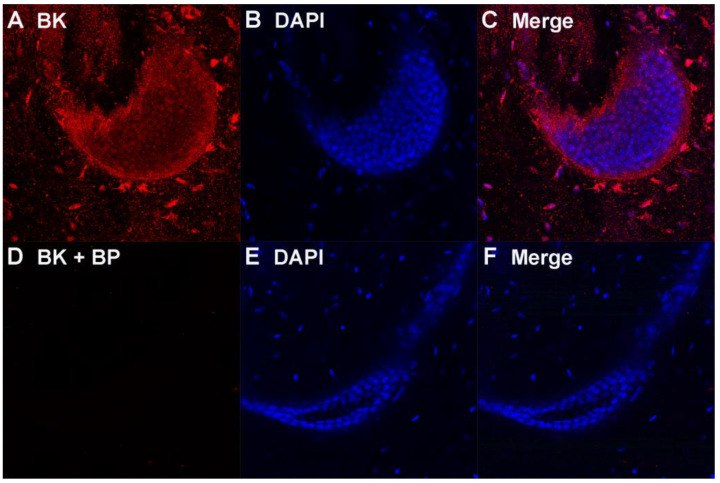
BKα immunoreactivity is specifically blocked by peptide immunogen. (**A**) anti-BKα KCNMA1 PA1-923 rabbit IgG, (**B**) DAPI nuclear stain, (**C**) merge of (**A**,**B**), (**D**) BKα + blocking peptide (BP), (**E**) DAPI and (**F**) merge of (**D**,**E**).

**Figure 2 cells-12-02125-f002:**
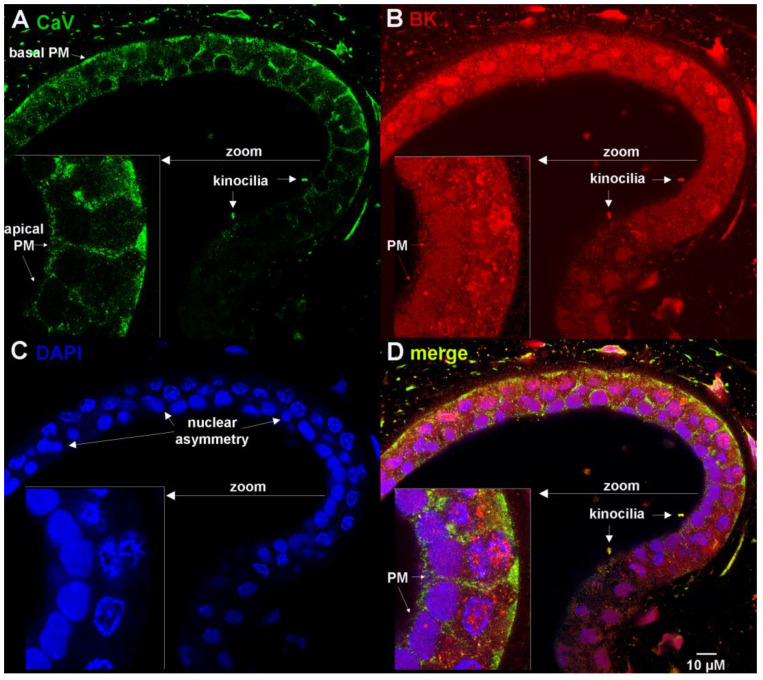
BK and CaV channel expression in plasma membranes, nuclei and kinocilia of electrosensory cells of ampullae of Lorenzini. Confocal section of alveolar wall co-immunostained with (**A**) mouse anti-CaV1.3 (z-stack in Appendix A), (**B**) rabbit anti-BK (z-stack in Appendix A) and (**C**) DNA counterstained by DAPI (z-stack in Appendix A). (**D**) Merge of (**A**–**C**) (z-stack in Appendix A). PM, plasma membrane.

**Figure 3 cells-12-02125-f003:**
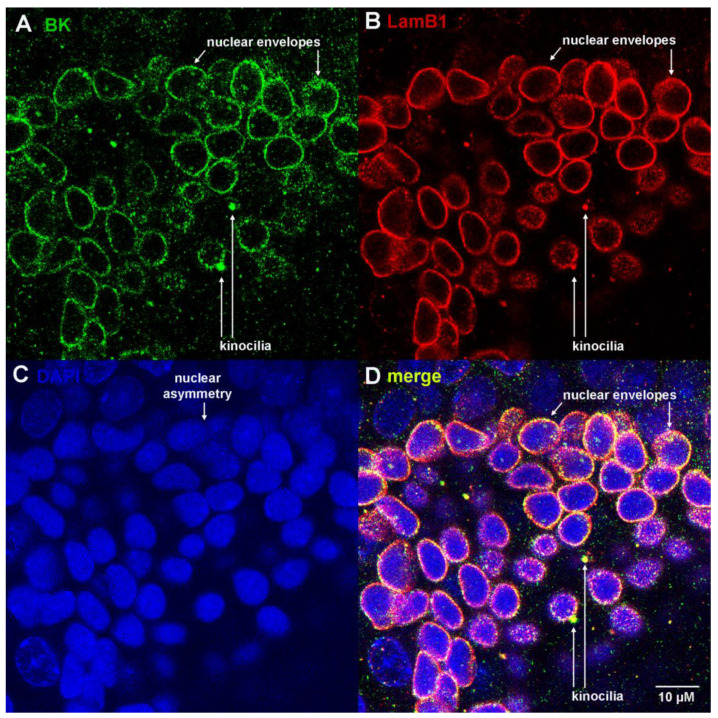
BK and lamin B1 are co-expressed in nuclear envelopes and kinocilia of electrosensory cells of ampullae of Lorenzini. (**A**) BK (mouse monoclonal MaxiKα), (**B**) lamin B1 (rabbit polyclonal) marks nuclear envelopes, (**C**) DAPI and (**D**) merge of (**A**–**C**) (z-stack in Appendix A).

**Figure 4 cells-12-02125-f004:**
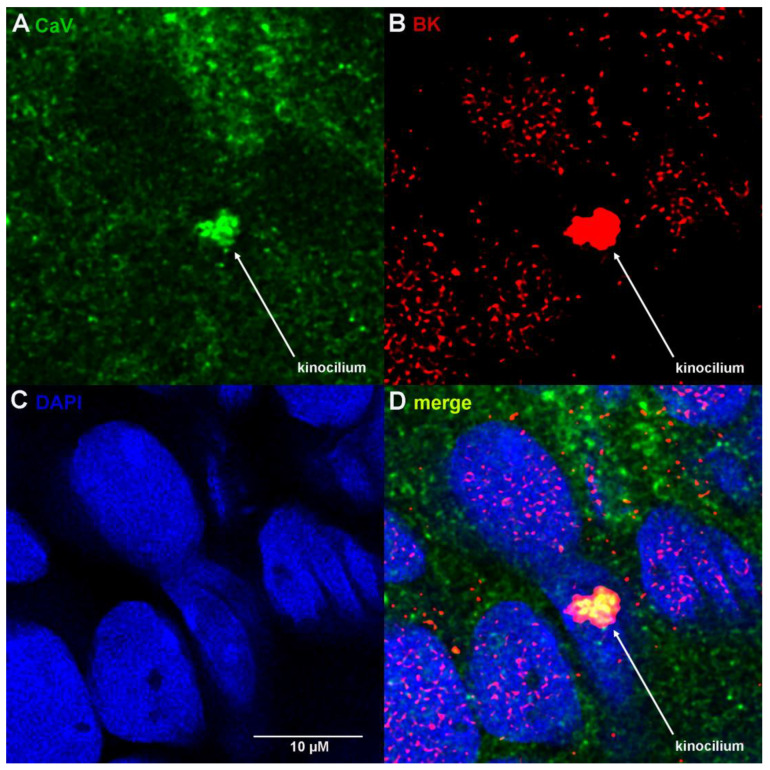
BK and CaV1.3 channels are co-expressed in sensory kinocilium. (**A**) CaV1.3, (**B**) BK, (**C**) DAPI and (**D**) merge of (**A**–**C**) (z-stack in Appendix A).

**Figure 5 cells-12-02125-f005:**
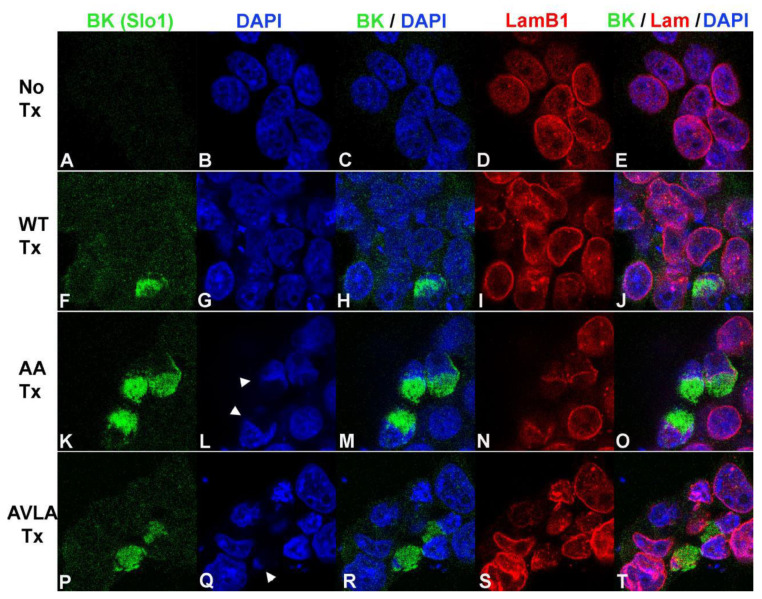
Localization of transfected skate BK channels and lamin B1 in HEK293 cells. (**A**–**E**): No transfection. (**A**) anti-BK (Slo1 mouse monoclonal). (**B**) DAPI. (**C**) Merge of (**A**,**B**). (**D**) Lamin B1. (**E**) Merge of (**A**,**B**,**D**). (**F**–**J**): WT transfection. (**F**) anti-BK (Slo1). (**G**) DAPI. (**H**) Merge of (**F**,**G**). (**I**) Lamin B1 (rabbit polyclonal). (**J**) Merge of (**F**,**G**,**I**). (**K**–**O**): AA mutation transfection. (**K**) anti-BK (Slo1). (**L**) DAPI. (**M**) Merge of (**K**,**L**). (**N**) Lamin B1. (**O**) Merge of (**K**,**L**,**N**). (**P**–**T**): AVLA mutation transfection. (**P**) anti-BK (Slo1). (**Q**) DAPI. (**R**) Merge of (**P**,**Q**). (**S**) Lamin B1. (**T**) Merge of (**P**,**Q**,**S**). Arrowheads in (**L**,**Q**) identify indentations of nuclei adjacent to accumulations of BK channels excluded from nuclei.

**Figure 6 cells-12-02125-f006:**
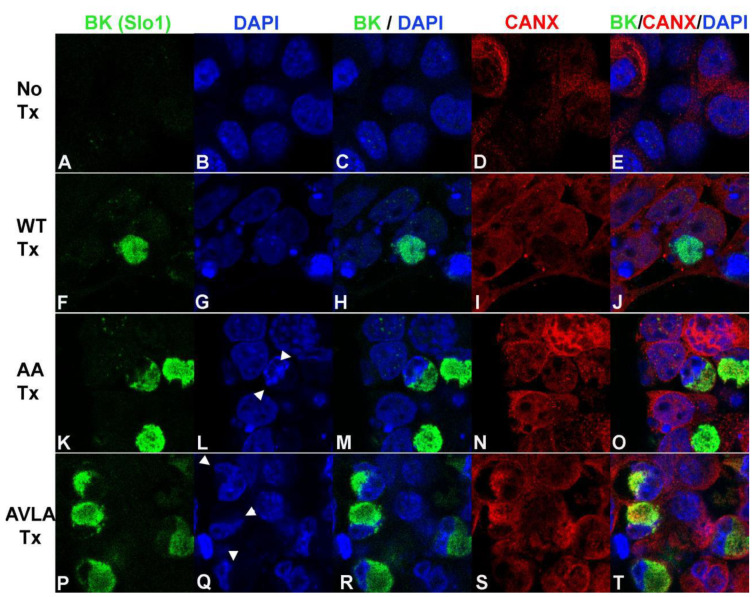
Localization of transfected skate BK channels and calnexin in HEK293 cells. (**A**–**E**): No transfection. (**A**) anti-BKα (Slo1). (**B**) DAPI. (**C**) Merge (**A**,**B**). (**D**) Calnexin. (**E**) Merge of (**A**,**B**,**D**). (**F**–**J**): WT transfection. (**F**) anti-BK (Slo1, mouse). (**G**) DAPI. (**H**) Merge of (**F**,**G**). (**I**) Calnexin (rabbit polyclonal). (**J**) Merge of (**F**,**G**,**I**). (**K**–**O**): AA mutation transfection. (**K**) anti-BK (Slo1). (**L**) DAPI. (**M**) Merge of (**K**,**L**). (**N**) Calnexin. (**O**) Merge of (**K**,**L**,**N**). (**P**–**T**): AVLA mutation transfection. (**P**) anti-BK (Slo1). (**Q**) DAPI, (**R**) Merge of (**P**,**Q**). (**S**) Calnexin. (**T**) Merge of (**P**,**Q**,**S**). Arrowheads in (**L**,**Q**) identify indentations of nuclei adjacent to accumulations of BK channels excluded from nuclei.

**Figure 7 cells-12-02125-f007:**
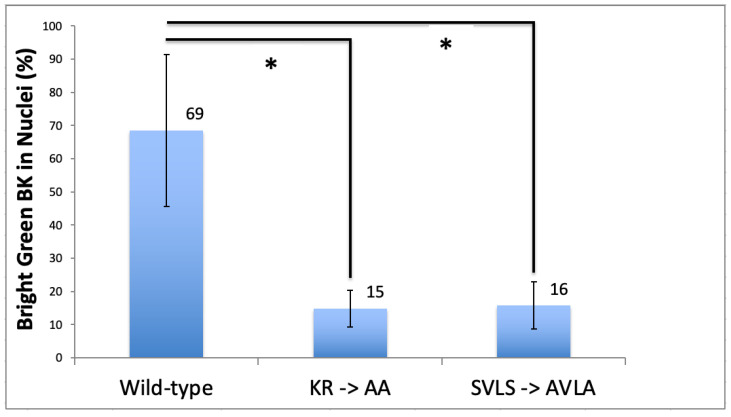
Localization of transfected skate BK channels in HEK293 cells. Significance of the differences between Wild Type, AA or AVLA mutation transfections was determined using the Kruskal–Wallis statistic. The difference between Wild Type and AA or Wild Type and AVLA mutations were significant at *p* < 0.05 (*).

**Table 1 cells-12-02125-t001:** Bipartite nuclear localization signal subject to alternative splicing.

King, KJ756351	KRIKKCGCKRL * * * * * * * * * * *QDENPSVLSPKKKQRNG
Bellono, KY355737	KRIKKCGCKRPRYGYNGYLSTIQDENPSVLSPKKKQRNG

## Data Availability

The data presented in this study are openly available in FigShare at the links provided in Appendix A.

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
