# Peer review of "Calcium-Activated Big-Conductance (BK) Potassium Channels Traffic through Nuclear Envelopes into Kinocilia in Ray Electrosensory Cells"

_cells, 2023, doi:10.3390/cells12172125_

Round 1
Reviewer 1 Report
Calcium-activated big conductance (BK) potassium channels traffic through nuclear envelopes into kinocilia in skate electrosensory cells by Chen et al. investigate the localisation of BK potassium channels, CaV1.3 calcium channels and the lamin B protein. They used mostly confocal imaging technics and some gene engineering to explore the different mutations of the BK genes and its affect to the localisation of the protein. The work is nice, clear and understandable. I have some comments and suggestion to the presentation of the data.
Minor comments:
1) line 26-27: keywords are missing.
2) line 73: species name should be in italic and should use the whole name in first time (with the description year and name of the scientist).
3) line 134: why ribbon synapses are mentioned here? These cannot be seen, and the authors did not work or analyse them later. Not even helps the reader to see clearer the figures. I do not even think the authors could have seen them, they know there should be, but could not approve it…
4) lines 198-199: the sequences could be put in a table, under each other to make the comparison easier.
5) CaV1.3 vs Cav1.3 - both are in the text.
6) They could mention what is lamin B. It could be figured out but would be easier if they would write it simply.
Major comments:
1) Materials and Methods misses the statistical analysis, and software which were used to it.
2) Figure 1: the insets should have a border. In lower magnification the zoomed parts should be bordered, to see it clearer. A kinocilia would look like nice to be zoomed. The abbreviations are not included in the legends.
3) Why the authors mention the ampullae wall is a single layer of cells? On the figures it seems to be multilayer – has at least 2-3 rows of nuclei. Is it pseudostratified?
4) Figures’ legends could be more descriptive.

Author Response
Minor comments:
1) line 26-27: keywords now added
2) line 73: species name should be in italic and should use the whole name in first time (with the description year and name of the scientist).
We now present the species name (Leucoraja erinacea, Mitchill, 1825), new line 72.
3) line 134: why ribbon synapses are mentioned here?
We eliminated this sentence about ribbon synapses, as suggested. New line 139.
4) lines 198-199: the sequences could be put in a table, under each other to make comparison easier.
We followed this suggestion and present new Table. New line 221.
5) CaV1.3 vs Cav1.3 – both are in the text.
We corrected each instance to CaV1.3 in the text, including References.
6) They could mention what is lamin B.
We identify that lamin B1 marks the inner nuclear envelope. New line 173.
Major comments:
1) Materials and Methods misses the statistical analysis.
We added that Microsoft Excel was used for statistical analysis. New line 132-133.
2) Figure 1: the insets should have a border… the zoomed parts should be bordered to see in clearer. A kinocilia would like nice to be zoomed. The abbreviations are not included in the legends.
This figure is now Figure 2. We added borders to the zoomed portions. We explain the abbreviation of PM to be plasma membrane in the legend.
The best zoom image of a kinocilium in cross-section in presented in new Figure 4.
3) Why the authors mention the ampullae wall is a single layer of cells? On the figures it seems to be multilayer – has a least 2 rows of nuclei. Is it pseudostratified?
We removed the “single-cell thick” description from the legend of Figure 2 to de-emphasize this point. The ampullae of Lorenzini wall was previously described as single-cell thick from electron microscopy studies, albeit from a different species, Aptychotrema rostrata (Wueringer, 2009). In Figure 2A, the CaV staining outlines the plasma membrane and is consistent with a conical shaped single cell spanning the wall. We were long perplexed by the appearance of two rows of nuclei in the wall image and carefully studied the confocal z-stacks of DAPI-stained nuclei. We determined that the nuclei exhibit morphology of asymmetric twisted dumbbells. This nuclear asymmetry is identified by arrows in Figure 2C and best appreciated in S3 movie, https://doi.org/10.6084/m9.figshare.19492001.v1
I imagine that the ampulla electrosensory cells might be organized like overlapping bristles of a paintbrush pushed sideways. This would be consistent with a pseudostratified morphology.
Reviewer 2 Report
The manuscript “Calcium-activated big conductance (BK) potassium channels 2 traffic through nuclear envelopes into kinocilia in skate elec-3 trosensory cells” provides the expression profiles of voltage-activated calcium channels (CaV1.3, cacna1d) and calcium-activated big conductance potassium channels (BK, kcnma1) in skate electrosensory cells. The author also identified the bipartite NLS sequence in skate BK channel and characterize their functions in intracellular trafficking of BK channels. This work is the first to descript the colocalization of CaV1.3 and BK in skate electrosensory cells for voltage gradient detection and to characterize the NLS sequences in skate BK complex. However, there are a few issues with the data presented in this manuscript.
First, the authors failed to validate the liability of the antibodies they used for immunostaining. Three different antibodies were used to probe BK channels (rabbit anti-BKα, MaxiKα, and mouse anti-Slo1). It remains untested whether those three antibodies recognize the same protein complex in skate electrosensory cells. If there are literatures showing the validity of those antibodies in immunostaining of skate tissues, those publications should be cited; if there is no previous evidence in research work with little skates, then the reactivities of those antibodies should be validated since none of the vendors claimed the reactivity in species other than human, mouse and rat.
Second, the claims that CaV1.3 and BK colocalize in the kinocilia is no convincing. Since single kinocilium is one of the morphological characteristics of skate electrosensory cells, one should expect kinocilia for all the electrosensory cells in both cross sections and surface preps. However, the microscope pictures presented in this manuscript only show kinocilia for a few cells, making readers question whether they are real kinocilia or just staining artifact. Kinocilium staining with specific antibodies should be performed to support such claim.
Another small issue is that the authors cited their bioRxiv paper in which some of the figures are identical to the figures in this manuscript. May consider removing such citation.
Author Response
First, the authors fail to validate the liability of the antibodies they used for immunostaining.
We present new data in new Figure 1 to validate the specificity of antibody PA1-923 used for immunostaining of BKalpha in skate ampulla. Here, we used a custom-synthesized peptide corresponding to the immunizing epitope for PA1-923. Pre-incubation of ampullae with blocking peptide at 25-molar excess over PA1-923 antibody, completely blocked BKalpha immunostaining (Figure 1D vs 1A). We conclude that the PA1-923 immunostaining in skate ampullae represents specific expression of BKalpha protein.
Second, the claims that CaV1.3 and BK colocalize in the kinocilia is no convincing.
We present movies of confocal z-stacks through the ampullae that better convey the colocalization of BK and CaV in rod-like structures which we interpret to represent kinocilia. This may be best appreciated in S5 Movie: https://doi.org/10.6084/m9.figshare.19492025.v1, at 8 seconds (near the bottom, 2 kinocilia) and 14 seconds (at the top left, 6 kinocilia), shown as yellow rods in transverse or cross-section, respectively.
We acknowledge that staining of kinocilia with alternative markers than BK, CaV and lamin B1 would be desirable. Our staining experiment with alpha-tubulin antibody was inconclusive because the ampullas we had available had to be unfrozen and rehydrated, and we are recently unable to procure fresh adult ampulla from Woods Hole Marine Biological Laboratories.
The authors cited their BioRxiv paper in which some of the figures are identical to the figures in this manuscript.
We eliminated the reference to BioRxiv preprint as suggested.
Round 2
Reviewer 2 Report
All my concerns are addressed. Just one miner issue, the interpretation of supplemental movies for cilium colocalization should be included in the main text.
Author Response
We now include the interpretation of supplemental movies for cilium colocalization in the main text, as requested: new lines 187 - 202; also in the supplemental movie descriptions: new lines 408-409 and 414-415.